# Increasing Solvent Tolerance to Improve Microbial Production of Alcohols, Terpenoids and Aromatics

**DOI:** 10.3390/microorganisms9020249

**Published:** 2021-01-26

**Authors:** Thomas Schalck, Bram Van den Bergh, Jan Michiels

**Affiliations:** 1VIB Center for Microbiology, Flanders Institute for Biotechnology, B-3001 Leuven, Belgium; thomas.schalck@kuleuven.be (T.S.); bram.vandenbergh@kuleuven.be (B.V.d.B.); 2Centre of Microbial and Plant Genetics, KU Leuven, Kasteelpark Arenberg 20, B-3001 Leuven, Belgium

**Keywords:** product toxicity, stress-response pathways, solvent tolerance, yeast, bacteria, bioproduction, fermentation

## Abstract

Fuels and polymer precursors are widely used in daily life and in many industrial processes. Although these compounds are mainly derived from petrol, bacteria and yeast can produce them in an environment-friendly way. However, these molecules exhibit toxic solvent properties and reduce cell viability of the microbial producer which inevitably impedes high product titers. Hence, studying how product accumulation affects microbes and understanding how microbial adaptive responses counteract these harmful defects helps to maximize yields. Here, we specifically focus on the mode of toxicity of industry-relevant alcohols, terpenoids and aromatics and the associated stress-response mechanisms, encountered in several relevant bacterial and yeast producers. In practice, integrating heterologous defense mechanisms, overexpressing native stress responses or triggering multiple protection pathways by modifying the transcription machinery or small RNAs (sRNAs) are suitable strategies to improve solvent tolerance. Therefore, tolerance engineering, in combination with metabolic pathway optimization, shows high potential in developing superior microbial producers.

## 1. Introduction

For years, biosynthetic pathways of various microorganisms have been exploited and optimized to produce valuable biochemicals and fuel molecules with solvent properties [1,2]. Recent concerns about crude oil availability and climate change further encourages the use of microbes in synthesizing solvent chemicals [3]. Compared to petrochemical production, the microbial approach requires inexpensive growth substrates and occurs at energetically favorable conditions [4,5]. In addition, these biochemical processes often rely on (plant-based) sugars, instead of ancient carbon storages, which is beneficial from a “closed carbon cycle” perspective. However, producing these fuels and biochemicals in a renewable microbial-based setup brings its own challenges. First, wild yeasts and lactic acid bacteria might contaminate the fermentation vessel, which causes significant production losses [6]. Secondly, the microbial producers are exposed to multiple stresses during the fermentation process including extreme fluctuations in (sugar) osmolarity, pH, temperature, and oxygen [7,8]. While the previous challenges are largely dependent on the process parameters, end-product toxicity is inherently linked to the production of fuels and biochemicals and becomes more dominant towards the end of the fermentation cycle [9]. Indeed, accumulation of these solvent molecules damages key cellular components, such as membranes, and interferes with enzyme function and energy metabolism [10,11]. As a result, cell growth ceases, production hampers, and eventually cell death increases when product concentration reaches lethal levels.

Improving tolerance to alcohols in *Clostridium* and in *Saccharomyces cerevisiae* was previously found effective to overcome solvent toxicity and to reach higher end-product concentrations [12,13]. Recently, researchers started to focus on non-model producers (e.g., *Corynebacterium glutamicum*) and less conventional chemicals (e.g., styrene) which resulted in more industry-relevant strains by manipulating stress-response pathways [14,15]. These examples illustrate that microbial tolerance engineering, along with metabolic engineering, is relevant in reaching economically attractive titers of alcohols, aromatics and more complex terpenoids in a variety of microbial species (Table 1). Although these solvent molecules differ in terms of chemical properties, this review focuses on the common “mode-of-toxicity” of these compounds. Furthermore, we compare the corresponding stress responses across various production hosts to identify overlapping tolerance strategies that are therefore widely applicable as a means to increase production.

## 2. Production of Solvent Molecules Using Microorganisms

Microbes can synthesize alcohols through various pathways, resulting in alkanols with different properties [1,31]. The most common one, ethanol, is traditionally produced by yeast (*S. cerevisiae*) or the bacterium *Zymomonas mobilis* to create alcoholic beverages (e.g., beer, wine and fruit-based spirits) or bioethanol fuels through the well-known Embden–Meyerhoff–Parnas (EMP) or Entner–Doudoroff (ED) fermentation routes (Figure 1) [27,32]. To obtain more complex alkanols such as (iso)propanol, (iso)butanol or long-chain fatty alcohols, researchers have adapted fatty acid synthesis, keto acid, and isoprenoid pathways of *E. coli*, *Pseudomonas*, *Clostridia* and *Ralstonia* species [33,34]. In this way, specific branched, long-chain or unsaturated alcohols are obtained that can be applied as plasticizers, polymer precursors or fuels [35,36,37,38,39].

In the last decade, researchers also successfully adapted microbial metabolism for terpenoid production. These compounds originate from the central five-carbon isoprene precursor and are either purely aliphatic (i.e., reduced) or decorated with ketone, alcohols or ether groups (i.e., oxidized) [40]. Biochemically, terpenoids are derived from either the mevalonate (MEV) or 1-deoxy-xylulose 5-phosphate (DXP) pathway (Figure 1). From an industrial perspective, *E. coli*, as well as *S. cerevisiae* (less commonly), is generally the preferred organism since optimization of the native DXP pathway results in high titers of isopentenols [41]. For monoterpenes (e.g., pinene, camphene, and limonene) and sesquiterpenes (e.g., farnesene and bisabolene), the microbial metabolism falls short and additional plant-based enzymes are heterologously expressed to improve yields [40]. The majority of terpenoids are implemented as high-energy (jet)fuels, but some molecules also serve as pharmaceuticals (e.g., Artemisinin as an antimalarial drug) [42].

Aromatics constitute the last class of biosynthetic molecules discussed in this review and are mostly used in the polymer and resin industry where benzene-, toluene- and xylene- (BTX) derivatives play a pivotal role [43,44,45]. The attractive application potential has increased interest in more cost-effective microbial production routes for these petrol-derived compounds [46]. These BTX derivatives originate from the phenylalanine and tyrosine amino acid biosynthetic pathways and examples thereof include phenol, *p*-hydroxystyrene, and *p*-hydroxybenzoate, which are preferentially produced in *Pseudomonas putida* [47,48] (Figure 1).

## 3. The Primary Cell Components and Processes Impacted by Solvent Toxicity

Fuel molecules and biochemicals typically affect multiple cell components and functions. This section comprehensively summarizes the recurrent “mode-of-toxicity” linked to alcohols, terpenoids, and aromatics (Figure 2).

### 3.1. Solvents Disrupt Cell Envelope Integrity

The cell envelope is a multilayered structure that protects against the (hostile) environment. In bacteria, it comprises the cytoplasmic membrane (CM), the peptidoglycan cell wall (CW) and, in case of Gram-negatives, an additional outer membrane (OM), whereas the cell envelope in yeast consists of a plasma membrane and a chitin-rich CW (Figure 2) [53,54]. As the primary interface between a microbe and fuel or solvent molecules, the cell envelope has been studied intensively. Various chemical compounds were identified to profoundly impact the membrane phospholipid composition of *E. coli*, but in general, the toxicity of a solvent highly correlates with its hydrophobicity [10,55,56]. Ethanol, for example, is a relatively hydrophilic alcohol due to its short two-carbon aliphatic tail and its polar hydroxyl-group. As a consequence, ethanol preferentially accumulates at the lipid/water interface of membranes—its methyl group pointing towards the hydrophobic core—and interacts with the polar phosphate groups of the phospholipids through hydrogen bonds [57,58]. Starting from 1 mol%, the partitioning of ethanol at the lipid/water boundary progressively expands the bilayer surface due to an increase in area per lipid and reduces the membrane thickness (Figure 3) [59]. Consequently, key physical properties of the CM are radically altered and cause an increase in permeability, fluidity, and disorder and a drop in surface tension and rigidity [58,60]. As such, the highly ordered, ethanol-free crystalline phase (*L_c_*) transitions towards a (partially) disordered gel phase (*L_β_*’) (Figure 3) [61,62]. In increasing ethanol concentrations (*ca.* 2.5 mol%), alcohol molecules start to penetrate deeper into the lipid bilayer, where they temporarily engage in hydrogen bonding with the lipid tails. Once ethanol has access to the membrane interior, a small fraction can cross the membrane barrier, ending up in the cytoplasm [58,59]. When the inner core of the CM is progressively enriched with ethanol molecules, the CM becomes more hydrophilic until the geometry of the bilayer turns into a compressed interdigitated state (*L_β_I*) (Figure 3).

In contrast to hydrophilic compounds, terpenoid molecules (e.g., limonene, farnesene/farnesol) display pronounced hydrophobic characteristics due to their overall bulky aliphatic structures. Terpenoids accumulate more rapidly into the hydrophobic core of the membrane compared to the small and more polar ethanol molecules (10^10^ enrichment of terpenoids in membrane vs. solution) and are therefore often applied as skin penetration enhancers for improved drug delivery [63,64]. Enrichment of these hydrophobic compounds in the interior of the bilayer causes membrane swelling and eventually ruptures the phospholipid barrier at extreme concentrations [64]. The discussed cases illustrate that the degree of hydrophobicity determines the partitioning of the solvent, either at the interface for hydrophilic molecules, such as short-chain alcohols, or in the inner core for more hydrophobic terpenoids. Of course, molecules that share physicochemical properties with both categories (e.g., the more polar oxidized terpenoids, long-chain alcohols, and phenolic molecules) show mixed behavior in terms of membrane partitioning.

In contrast to the CM-associated effects, the impact of solvents on the CW and the OM is less documented and the molecular interaction between solvents and these envelope layers remains unclear. As an exception, limonene solely targets the CW of yeast, and not the CM, at ca. 790 µM [65]. In case of other chemical species, the effect of solvents on the CW is less apparent and more indirect. For example, reduced peptidoglycan crosslinking in *E. coli* results from an ethanol-mediated downregulation of biosynthesis enzymes at 0.7–0.9 M rather than as a consequence of a direct solvent–CW interaction [66,67]. In *Clostridium* species, the cell wall undergoes morphological changes, including thinning, at the transition from the acid production to the solventogenic phases [68]. The OM is more rigid than the CM and acts as an effective permeability barrier to hydrophobic solvents due to the “gel-like interior” of lipopolysaccharides (LPSs). On the one hand, this property is attributed to the anchoring of several fatty acid chains per LPS molecule, which stably embeds each of the units into the OM. On the other hand, the high abundance of hydrogen and hydroxyl groups in lipid A promotes lateral interactions between LPS molecules through hydrogen bonds [69]. Despite its reinforced structure, ethanol may still impact the integrity of the OM as LPS leaches out of the OM under high ethanol concentrations (1.7 M) [70]. Interactions between ethanol and the OM lipids likely proceed similarly as is the case for the CM since the OM lipids are either identical to those in the CM (e.g., glycerophospholipids) or share the same hydrophobic-polar structure (e.g., lipid A in LPS) [71]. However, the overall OM structure is assumed to be less severely affected by alcohol due to the more impermeable and resistant nature of the OM.

Ultimately, solvent molecules not only disrupt the barrier, but a loss of CM integrity also promotes leakage of nutrients and ions. The latter results in a diminished proton motif force which, together with quinone malfunctioning, severely reduces ATP synthesis (Figure 2) [11,72,73].

In addition to lipids, proteins are also abundantly embedded in the CM and are often involved in energy generation and nutrient or ion transport processes. Inevitably, these CM-bound proteins will be affected by alcohol exposure [74,75]. On the one hand, ethanol can directly affect the function of these proteins (the “protein hypothesis”). On the other hand, this alcohol could primarily target the bilayer structure and change the physicochemical properties of the membrane to indirectly disturb membrane-associated proteins (the “lipid hypothesis”) [76]. Or, both phenomena could occur at the same time. Although most evidence comes from studying the influence of alcohol on neuronal ion channels in the context of alcohol abuse and anesthetics, the same principles might also (partially) explain general solvent toxicity on (ion-transport) channels and membrane-bound enzymes in microorganisms.

### 3.2. Accumulation of ROS and Radicals during Solvent Stress Damage Biomolecules Inside the Cell

Reactive oxygen species (ROS) are a major cause of microbial cell death under several stress conditions, including antimicrobial treatments [77,78]. Additionally, in the case of solvent stress, ROS and toxic radicals tend to accumulate and affect lipids, proteins, and nucleic acids. This paragraph discusses how solvent molecules induce these adverse chemical species that are key in the secondary effects of solvent exposure (Figure 2).

First, ROS mostly originate from electron leakage at the electron transport chain (ETC) or are derived from P450 cytochromes as a result of alcohol abuse in humans or during fermentation in microorganisms [79,80,81,82]. Some ROS, such as hydrogen peroxide, produce hydroxyl radicals by the spontaneous Fenton reaction in the presence of free ferric ions or at iron–sulfur clusters of certain proteins [66,83,84]. These hydroxyl radicals are highly reactive and peroxidate lipids, damage DNA and proteins and convert ethanol into a 1-hydroxyethyl radical, which is also deleterious to proteins and antioxidants [85,86]. Although the previously described ROS cascade is accepted as major cause of solvent-mediated ROS damage, Burphan et al. recently discovered a still poorly understood, mitochondria-independent ROS pathway when yeast was exposed to 1.7 M ethanol [87]. Apart from ethanol, exposure to 1.2 mM limonene also elicits a burst of ROS in yeast [88].

Secondly, some microbial species can catabolize aromatics to detoxify these undesired chemicals and, simultaneously, replenish cellular energy. However, such degradation through the oxidative metabolism might also accumulate ROS. A well-known example in some *Pseudomonas* species is the bioconversion of benzene or toluene into catechol intermediates [89,90]. These transition products are prone to heavy metal-involved oxidation which results in semiquinone radicals that can form stable DNA adducts or cross-link to sulfhydryl groups of proteins [91]. For catechol-like terpenes, such as diterpenone catechol, the same radical chemistry and damaging effect is directly applicable without the need of a primary bioconversion step [92].

### 3.3. Solvents Damage DNA and Impede Transcription and Translation Processes

The presence of solvent-induced ROS accumulation creates a hostile (intracellular) environment for biomolecules such as nucleic acids. Indeed, research in yeast and *E. coli* has demonstrated that, at 0.85 M ethanol, lethal DNA lesions in the form of single-strand and double-strand breaks (SSBs and DSBs) start to appear (Figure 2) [83,93]. Additionally, ROS-oxidized nucleotide bases (e.g., 8-oxo-deoxyguanosine) are frequently incorporated into the genome. Subsequent incomplete base-excision repair of these closely spaced, aberrant bases may result in a lethal DSB [94,95]. Moreover, the formation of stable DNA adducts contributes to the mutagenic character of solvent molecules [96]. As mentioned earlier, the benzene- and toluene-derived semiquinone radicals give rise to these mutagenic DNA adducts, but malondialdehyde and 4-hydroxynonenal—two lipid peroxidation products—and ethanol-derived acetaldehyde also share this ability [81,91].

In addition to causing radical and ROS-mediated DNA damage, solvents also have an influence on genome stability. Ethanol interferes with the replisome which results in replication fork stalling (Figure 2). In turn, this replication defect recruits translesion polymerases which inherently display higher error rates and increase the mutation rate [97]. Furthermore, ethanol inhibits cell cycle progression in yeast by disrupting spatial organization of actin and a similar cell-cycle arrest was also observed in *Candida albicans* terpenoid-treated cells [98,99].

In addition to harming the integrity of nucleic acids, Haft et al. showed that ethanol, in the range of 0.85 to 1.4 M, dramatically affects transcription and translation processes in *E. coli* (Figure 2) [100]. Indeed, ethanol is responsible for increased ribosome stalling and aberrant termination which uncouples translation from transcription. The latter, together with an ethanol-induced decrease in RNA polymerase (RNAP) activity, renders transcription more susceptible for Rho-dependent termination. Besides perturbing transcription-translation coupling, ethanol also stimulates translational misreading which gives rise to a pool of error-prone proteins.

### 3.4. Solvents Affect the Structure and Function of Proteins

The cell’s arsenal of functional proteins and enzymes will eventually be reduced by mistranslation and disruption of de novo protein synthesis under solvent stress. However, solvents also have a direct impact on the existing protein pool (Figure 2). Solvents disturb the polarity of aqueous media and therefore weaken hydrophobic interactions that assist proteins to fold into their native structures [101,102]. Disruption of these stabilizing bonds not only results in a collapse of the native protein structure, but also alters (re)folding thermodynamics [102,103]. Although destroying protein structure has detrimental repercussions for cell functioning, complete denaturation of polypeptides only occurs at extreme ethanol concentrations, usually exceeding 3.4 M [104,105]. Hence, the relevance of alcohol-induced protein unfolding is debatable in microbial fermentation settings. At lower concentrations (0.85–1.7 M), however, ethanol might directly bind to transcription factors (TFs) and inhibit glycolytic enzymes of *Z. mobilis* and *S. cerevisiae* [66,106,107]. This enzyme malfunctioning is often attributed to secondary effects, related to a loss of membrane integrity upon ethanol exposure. Indeed, a drastic change in the membrane lipid environment is known to affect proper functioning of membrane-bound proteins, such as ATPases [108]. The latter impedes replenishing of cellular energy and, as a consequence, hampers ethanol production and other cellular processes [109]. Finally, proteins are also damaged by ethanol-induced ROS that oxidize sensitive amino acid residues, disulfide bonds, and iron–sulfur clusters. As a result, these extensively oxidized proteins are prone to cross-linking and aggregation, complicating their degradation [66,110].

## 4. Adaptive Responses Protect Cells from Solvent Exposure

Solvents impose a multifaceted stress on fermentative microorganisms. Hence, microbes adapt their cellular processes and fine-tune the composition of cellular components to overcome the solvent-induced stress and improve their fitness. The adaptation mechanisms have been intensively studied using laboratory-based evolution experiments (for a list of conducted experiments see CAMEL: https://cameldatabase.com [111]) and omics-driven approaches (for further reading, see [112]). This section highlights the most common aspects in tolerance development among various microbial species and against chemically distinct solvents.

### 4.1. Increased Mutation Rates Accelerate Solvent Adaptation

Solvents delay replication forks and damage genomes in a ROS-dependent manner which upregulate error-prone DNA polymerases [97,113]. Since these rescue systems lack proofreading activity, they can evoke secondary mutations in mismatch repair enzymes, thereby increasing the mutation rate. Indeed, this hypermutation phenotype is commonly found in (long-term) evolution experiments under ethanol stress [66,114,115,116]. Although the accelerated mutation frequency is not causally implicated in solvent tolerance as such, this (dynamic) hypermutation phenotype increases the chance of driver mutations emerging, thereby speeding up the adaptation process [115].

### 4.2. Maintaining Cell Envelope Integrity to Overcome Solvent Stress

Since the cell envelope is the primary barrier between the cell and its toxic environment, adaptation mechanisms focused on this cellular component have been the subject of numerous studies. The vast majority deals with CM fatty acid modifications to ensure that the membrane’s optimal fluidity is maintained, a process called homeoviscous adaptation [10]. Carey and Ingram noticed that the *Z. mobilis* bacterium, which can tolerate ethanol concentrations up to 2.17 M, intrinsically has high *cis*-vaccenic acid (Δ^11^Z-C18:1) levels in its membrane, suggesting a possible relationship between long-chain unsaturated fatty acids and ethanol tolerance [117,118]. Additionally, *E. coli* and the yeasts *S. cerevisiae* and *K. marxianus* respond to ethanol (0.69-2 M) by increasing the proportion of *cis*-vaccenic acid (Δ^11^Z-C18:1) or oleic acid (Δ^9^Z-C18:1), respectively, at the expense of palmitic acid (C16:0) [119,120,121,122]. Indeed, further evidence from laboratory evolution-based experiments highlights that a high unsaturated fatty acid (UFA):saturated fatty acid (SFA) ratio and an overrepresentation of long-chain fatty acids are key for ethanol tolerance development [123]. At first sight, replacing the more rigid saturated FAs with their more disordered and unsaturated analogs might seem counterintuitive in the presence of a fluidizing solvent such as ethanol. However, this adaptation mechanism provides a way to (partially) restore the bilayer geometry (Figure 3). Furthermore, the response to higher alcohols, such as *n*-butanol, is more diverse and seem to be more species-specific. For example, enrichment of UFAs protects *E. coli* against *i*- or *n*-butanol, but in the native butanol producer, *Clostridium acetobutylicum*, the opposite is true at around 0.15 M [124,125,126]. In other bacterial species, including *O. oeni*, adjusting membrane fluidity under alcohol is accomplished by increasing the level of cyclopropane fatty acids (CFAs) instead of saturated FAs [127,128]. These unusual FAs might display a SFA–UFA chimeric effect, maintaining both membrane fluidity and improving chain ordering of lipid tails because the cyclopropane moiety restricts rotational motion of neighboring FA bonds [129]. In the case of tolerance to aromatics (28–95 mM), *P. putida* exploits the fast-responsive cyclopropane FA synthases and *cis*-*trans* isomerases to optimize lipid tail packing and ordering of preexisting FAs [130,131].

In addition to modifications restricted to the aliphatic lipid tails, polar head groups of phospholipids also influence surface charge, polarity, and membrane thickness [10]. Hence, microorganisms adjust the fraction of each phospholipid (such as phosphatidylserine, -ethanolamine, -choline, -glycerol, and cardiolipin) in response to toxic solvents. However, research results are not always consistent on the changes in phospholipid composition even within the same organism and under the same chemical stressor. For instance, research pointed out that the level of zwitterionic phosphatidylethanolamine generally decreases in *Z. mobilis* and *E. coli* when exposed to ethanol stress (between 0.2 and 1 M) [117,132]. Since the anionic/zwitterionic head group ratio would consequently increase, researchers assumed that optimizing electrostatic repulsion between neighboring phospholipids might be crucial for improving solvent tolerance [10]. However, more recently and contrary to previous findings, it was shown that phosphatidylethanolamine is responsible for membrane thickening which confers tolerance to a series of (non)alcoholic and aromatic compounds as well as to lignocellulosic inhibitors [133]. Moreover, the phospholipid response is sometimes highly strain-specific. As an illustration, *P. putida* Idaho increases phosphatidylethanolamine levels whereas in the DOT-T1E strain, a rise in the cardiolipin fraction is most noticeable [134,135]. This particular enrichment of cardiolipin is probably beneficial for the function of Resistance–Nodulation–Cell Division (RND) superfamily efflux pumps rather than for stabilizing the membrane structure [136].

In addition to phospholipids, *S. cerevisiae*, *Yarrowia lipolytica* and *Z. mobilis* incorporate a significant amount of sterols or sterol-like molecules in their membranes. Aside from UFAs, these lipids prevent the membrane from shifting towards the interdigitated phase and consequently avoid membrane thinning (Figure 3) [62,137,138]. Hence, modifications of the fatty acid tail and the polar head group in combination with cyclic lipids (i.e., ergosterols and hopanoids) all contribute to maintaining membrane fluidity (*cf.* homeovisous adaptation), but also ensures that critical processes can proceed optimally, even under solvent stress. Apart from lipids, the microbial CM also consists of membrane-associated proteins. In fact, a nonspecific increase in membrane proteins rigidifies the CM and thus helps to counteract the fluidizing effect of ethanol [55,117].

The next protective barrier of the bacterial envelope is the peptidoglycan CW. Here, several studies have shown that an upregulation of peptidoglycan biosynthesis genes is associated with ethanol and butanol tolerance in *E. coli* and *L. plantarum* [139,140,141,142]. Additionally, *S. cerevisiae* extensively remodels its CW to acquire tolerance to 1.2 M ethanol and jet fuels (such as 790 µM limonene) [65,143,144].

Finally, the OM is the outermost layer of the cell envelope in Gram-negative bacteria, in which LPSs take up a significant proportion. The majority of literature agrees that upregulating LPS synthesis genes (such as *lpcA*) has a positive effect on ethanol and butanol tolerance in *E. coli* and xylene tolerance in *P. putida* Idaho [139,145,146,147,148]. An increase in LPS has been proposed to decrease the cell surface hydrophobicity which consequently prevents binding of organic solvent molecules [149]. In contrast, the cell surface of *P. putida* becomes more hydrophobic in the presence of octanol because this bacterium releases outer membrane vesicles, enriched with rather hydrophilic B-band LPSs [150,151]. This increased hydrophobicity has been linked to enhanced biofilm formation, a lifestyle which offers protection against a toxic environment [150,151]. Similar to the plasma membrane, the OM also consists of membrane proteins, such as porins [152]. Particularly, the EnvZ/OmpR sensor together with the downstream regulated OmpC and OmpF porins significantly determine OM permeability and bacterial survival under ethanol stress in *E. coli* [153,154]. Additionally, excluding the outer membrane Protein F in *Pseudomonas aeruginosa* improves toluene tolerance due to reduced solvent influx [155].

### 4.3. Adaptive Mutations Related to the Transcription and Translation Machinery Counter Solvent-Associated Aberrations

To counteract undesired transcription and translation defects, Haft et al. identified mutations in RpsQ and Rho of *E. coli* [100]. The former increases translation accuracy during protein synthesis and the latter reduces premature transcript termination as ethanol slows down the RNA polymerase. Moreover, ethanol-associated translation inhibition seems to be strongly located at nonstart AUG (methionine) codons, and therefore *E. coli* responds by adjusting methionine metabolism. Hence, deleting the methionine biosynthesis repressor or supplementing exogenous methionine protects *E. coli* against lethal ethanol stress. In the same way, overexpression of the methionine activator (*metR*) might also explain improved tolerance to 17 µM isopentenol [156].

### 4.4. Protein Folding and Chaperone Activity Restore Protein Function

Protein malfunction, often because of misfolding or aggregation, is one of the most severe survival-reducing and growth-limiting effects of solvents. Yeast cells induce a range of Heat-Shock Proteins (HSPs) and unfolded protein response (UPR) gene members that are involved in disaggregation of denatured proteins upon ethanol stress [157,158]. In addition, the ubiquitin-proteasome plays a crucial factor in butanol tolerance (at 0.13 M) in *S. cerevisiae*, indicating that turnover of damaged proteins is essential during solvent stress [159]. Additionally, in bacteria, heat-shock response-associated chaperones, such as DnaK, GroELS and ClpB, together with the GrpE nucleotide exchange factor play a universal role in alcohol and toluene tolerance [142,160,161,162]. Particularly in *O. oeni*, the heat-sensitive master response regulator, CtsR, is essential in ethanol tolerance (at 1.9 M) and controls expression of stress-responsive proteases (including ClpP) and the chaperones DnaK and GroESL [163,164]. Moreover, overexpression of these HSPs is often a minimal requirement for further butanol tolerance development in *C. acetobutylicum* and *E. coli* at 0.22 and 0.16 M, respectively. Indeed, when combined with transporter systems or FA-synthesis-related genes, HSPs potentiate their tolerance-improving properties [165,166].

### 4.5. Cell Metabolism Is Reprogrammed during Solvent Exposure

Solvent-related cell membrane damage is inevitably linked to reduced energy levels, since the proton motif force is impaired and ATP levels are consequently reduced [11,167]. Not surprisingly, energy restoring adaptation mechanisms have been identified in yeast and bacterial species exposed to ethanol, butanol, and toluene [145,168,169,170]. Interestingly, Cao et al. pointed out that cells, not adapted to 0.87 M ethanol, repress aerobic respiration-related genes and rely on alternative pathways (i.e., fermentation and β-oxidation) to generate cellular energy [66]. Indeed, Brynildsen and Liao confirmed that 0.11 M *i-*butanol causes quinones to dissociate from the membrane and that *E. coli* bypasses this defect by shutting down the TCA cycle and NADH dehydrogenases in an ArcA-mediated way. On the contrary, ethanol-adapted *E. coli* is capable of using aerobic respiration to replenish cellular energy more efficiently [66,145]. Similarly, in ethanol-tolerant *K. marxianus* strains, the aerobic TCA cycle is more active than in the parental strain, not adapted to alcohol [122]. Although the TCA cycle plays a pivotal role in energy generation, other pathways can contribute as well. For example, *L. plantarum* exploits its citrate metabolism to meet the energy needs under ethanol stress (at 1.37 M) [142]. Hence, striving to restore intracellular energy levels despite the presence of a pmf-disrupting solvent is a recurrent theme which is also applicable in case of butanol and toluene [171,172,173,174,175]. In yeast, ethanol toxicity can be diminished by supplementing potassium or by increasing the pH which helps to recover the electrical membrane potential that is needed for building up energy and driving transport processes [176].

Moreover, solvent tolerance has been linked to changes in uptake and metabolism of sugars and polyalcohols. In case of yeast, intracellular trehalose and inositol accumulation improves cell survival in high ethanol conditions (>2 M) [177,178,179]. On the one hand, trehalose is implicated in ethanol tolerance since this sugar is involved in membrane stabilization and conformational repair of denatured proteins [180,181]. On the other hand, supplementation of inositol increases the content of inositol-containing membrane lipids which reduces ion and nucleotide leakage and promotes H^+^-ATPase activity under ethanol stress [178]. In addition, galactose metabolism and sugar transport as well as glycerol accumulation also play a role in bacterial tolerance to ethanol, butanol, and toluene [140,145,173,182]. Particularly, mannose metabolism and its related phosphotransferase transporter system (PTS) (*manXYZ*) are often upregulated under solvent toxicity in *E. coli* [114,172,183]. This mannose PTS might change cell surface hydrophobicity which avoids influx of apolar *n*-hexane molecules (0.76 M) [184].

In addition, amino acid metabolism is extensively reprogrammed under ethanol stress because certain amino acids directly participate in stress-response mechanisms. In *E. coli*, upregulation of serine biosynthesis and glycine cleavage increases intracellular betaine levels, a well-known osmoprotectant, whereas proline, arginine and valine serve as stress-protectants in yeast [122,147,185,186]. In case of tolerance to butanol (*ca.* 85 mM), the osmoprotectants glutamate and alanine improve survival in *E. coli* [172]. In *C. acetobutylicum,* increased levels of branched amino acids are converted into branched fatty acids that help to optimize membrane fluidity under butanol stress (in a range of 54–270 mM) [182].

Finally, bioconversion of the end-product into a less toxic metabolite also reduces solvent stress. This strategy includes two microbial approaches: either (complete) degradation and consumption of the solvent or partial derivatization. First, end-product degradation is not attractive from an industrial point of view as the desired product cannot be recovered. However, this detoxification strategy is very favorable for the microbe itself, since breaking down solvent molecules simultaneously reduces toxin concentrations and enables the host to use the C-chain as an energy source. Indeed, Goodarzi et al. demonstrated that ^13^C-labeled ethanol ends up in the intermediates of the *E. coli* TCA cycle, supporting the detoxification hypothesis [186]. Moreover, research has revealed that alcohol/acetaldehyde dehydrogenases are upregulated in *E. coli* and that a mutant alcohol dehydrogenase in *Clostridium thermocellum* also confers ethanol tolerance (up to 0.87 M), suggesting that an altered alcohol metabolism might be an important factor in acquiring tolerance [114,187,188]. Particularly, *P. putida* exhibits high tolerance to aromatic hydrocarbons (e.g., toluene at 28 mM) due to the catabolic *tod* operon [189,190]. Secondly, end-product derivatization masks the toxicity of the original solvent by, e.g., adding sugars. Glycosylation of vanillin by an *Arabidopsis thaliana* glycosyltransferase in *Schizosaccharomyces pombe* or *S. cerevisiae* has been exploited to relieve solvent stress, thereby improving product yields [191,192,193]. In contrast to end-product degradation, recovery of the desired flagrance is feasible and the sugar moiety can be cleaved by glycosidases in postproduction processing steps [194].

### 4.6. Engaging Global Stress Responses Protects the Cell in the Presence of Solvent Molecules

Bacteria and yeasts exploit multidrug resistance (MDR) and oxidative stress-response mechanisms to acquire solvent tolerance. Apparently, the link between antibiotic resistance and solvent tolerance is bidirectional since fluoroquinolone-resistant clinical *E. coli* isolates display high tolerance to cyclohexane and, vice versa, *P. putida* cells, adapted to 6.5 mM toluene, are less susceptible to tetracycline and polymyxin [195,196]. Additionally, induction of the pleiotropic drug resistance pathway activates the PDR5 ABC transporter and confers tolerance to (cyclo)hexane and isooctane in *S. cerevisiae* [197]. The association between solvent tolerance and bacterial antibiotic resistance is attributed to global MDR regulators and efflux pumps. In *E. coli*, *marAB*, *soxRS,* and *rob* regulons contribute to the MDR phenotype because they upregulate the tripartite AcrAB-TolC multidrug efflux pump and reduce cell envelope permeability due to a decrease in porin expression [198]. Consequently, salicylate-mediated induction or reduced proteolytic degradation of *marA* as well as increases in *soxRS* and *robA* expression improve survival under cyclohexane stress (*ca.* 0.9 M) in an AcrAB-TolC-dependent way [149,199,200,201,202]. In particular, *Pseudomonas* species acquire tolerance to toluene (28 mM), xylene (81 mM), and *n*-hexane (152 mM) since the *P. putida* TtgABC and TtgGHI and *P. aeruginosa* Mex-Opr RND efflux systems are able to extrude (aromatic) hydrocarbons [203,204,205]. Moreover, heterologous overexpression of efflux pumps in *E. coli* is effective in terms of improving tolerance to the terpinoids limonene (around 2 mM) and pinene (in the range of 125–315 mM) [206,207,208]. Up until now, these cases confirmed the relevance of MDR efflux for tolerance to hydrophobic solvents. In the case of the more hydrophilic alcohols, the importance of efflux systems is more promiscuous. The ADP1 ATP-binding cassette (ABC) pump in *S. cerevisiae* and the TtgAB in *P. putida* play a role in tolerance to ethanol (up to 1.3 M) and higher alcohols (within 64–240 mM) [209,210]. However, the contribution of the *E. coli* AcrAB-TolC RND pump to alcohol tolerance is more unclear. Deleting the *acrAB* loci does not increase sensitivity of *E. coli* towards simple alcohols and only an artificial mutant AcrB is able to efficiently expel *n*-butanol when challenged with 76 mM [211,212]. These cases illustrate that alcohols do not belong to the native substrates of the AcrAB-TolC efflux pump. As such, the AcrAB complex is likely unable to recognize alcohols and hinders alcohol export since it unnecessarily occupies TolC. Consequently, TolC is less available for other efflux components (such as AcrD or EmrA), which are capable of recognizing isoprenol and therefore mutants, lacking *acrAB*, are less sensitive to alcohols [213]. In short, increasing solvent efflux can either be the dominant tolerance-improving adaptation mechanism, in the case of terpenoids, or might be less suitable in the case of simple alcohols.

As solvent stress is often accompanied by the accumulation of ROS, oxidative stress-response systems are generally upregulated during exposure to solvents. For example, high concentrations of alcohol in *E. coli* (0.85 M) and *O. oeni* (>2 M), toluene (0.94 M) in *P. putida,* and limonene (1.2 mM) in *S. cerevisiae* activate superoxide dismutases (*sod*), catalases (*katG*), and thioredoxins or glutathione reductases/peroxidases, the OxyR and SoxRS regulons and the SOS response, involved in DNA repair of ROS-induced damage [66,88,145,172,214,215]. Moreover, the link between oxidative stress and solvent tolerance in *S. cerevisiae* is even more pronounced as ethanol-induced ROS can serve as signal molecules to engage the ethanol stress-response system [216]. Here, superoxide ions are rapidly converted into hydrogen peroxide by the mitochondrial superoxide dismutase. In turn, hydrogen peroxide stimulates formation of disulfide bonds in the Yap1p transcription factor and as a result traps this protein inside the nucleus which enables Yap1p to trigger the ethanol defensive response. Although most microorganisms have the ability to deal with solvent-elicited oxidative stress, Chin et al. have also implemented heterologous expression of metallothioneins to scavenge ROS and thereby improved tolerance to ethanol (0.87 M) and butanol (164 mM) [217].

## 5. Engineering Microorganisms for Improved Tolerance and Production

Ideally, all gains in microbial tolerance towards industrial-relevant alcohols, terpenoids, and aromatics should ultimately be reflected by a higher product output. Intuitively, effective tolerance engineering programs should result in a higher fraction of viable or metabolically active cells that, in turn, could more efficiently participate in the production process. However, not all tolerance strategies will turn out successfully. For example, induction of CFA synthesis in *Clostridium* or serial adaptation in *E. coli* did not necessarily enhance ethanol and (iso)butanol production [139,218]. These case studies indicate that there is not always a one-to-one link between tolerance and production. Fortunately, there is also more promising evidence that microbial productivity can benefit from improved tolerance (Table 2).

Adaptive laboratory evolution is a powerful tool to improve industrial-relevant features of microbial producers or to study the evolutionary trajectory towards complex traits (Table 2) [2,111]. Not surprisingly, this approach has been applied to adapt yeast and bacteria to the toxic end-product in an attempt to increase solvent titers. In 1998, Yomano et al. evolved *E. coli* through serial cultivation under ethanol stress and isolated a tolerant clone which produced more ethanol than its ancestor [219]. In yeast, Thammasittirong et al. UV-mutagenized ethanol-adapted *S. cerevisiae* cells to further improve survival and production yields [220]. More recently, researchers successfully expanded the ALE approach to other microbial species and solvent molecules because these alternative producers possess attractive industrial characteristics (e.g., thermotolerance) or because the different end-products display more advantageous (fuel) properties (e.g., higher energy density upon combustion) [12,207,221,222,223]. For example, Wang et al. exploited methanol tolerance in *C. glutamicum* to enhance methanol bioconversion instead of solvent accumulation [14]. The resulted methylotrophy could therefore be applied to increase production of valuable L-glutamate on inexpensive (natural gas-based) methanol. In addition, improved ethanol tolerance even finds an application in the food industry where ethanol-adapted *O. oeni* strains reach a higher level of L-malic acid into L-lactic acid conversion and, hence, contribute to the microbial stability of (red) wines more efficiently [224]. Moreover, Lennen et al. demonstrated that evolved populations can acquire cross-compound tolerance to other alcohols, diols, diamines, etc. [222]. Indeed, a previous report also stated that overlapping stress responses are involved in the presence of distinct industrially relevant chemicals [225].

Increasing microbial tolerance levels by upregulating specific stress responses is another suitable approach to boost solvent production in the case of alcohols, terpenoids (e.g., limonene), and (hydroxylated) aromatics (e.g., phloroglucinol and vanillin) (Table 2). In the traditional ethanol producers, heterologous overexpression of a *Lactobacillus plantarum* peptidoglycan synthesis gene in *E. coli* or upregulation of a native ATP-binding efflux pump in *S. cerevisiae* resulted in significant production gains [141,210]. For higher alcohols and phloroglucinol, overcoming methionine limitation by upregulating the corresponding biosynthesis pathway and improving protein stability by activating chaperones contributed to increased yields [156,226,227]. In case of terpenoid molecules (e.g., limonene and amorphadiene), overexpressing efflux pumps in *E. coli* to expel toxic solvent molecules was highly effective to optimize product titers [206,208]. Lastly, introduction of glycosyltransferases in yeast species not only relieved vanillin toxicity but also resulted in higher product yields [191,192,193].

Then, the last two strategies are particularly suited to elicit a genome-wide, multi-pathway stress response. First, global transcription machinery engineering (gTME) allows the reprogramming of a species’ transcriptome by modifying a central gene with key transcription activity (Table 2). Specifically, a diverse mutant library of the target gene of interest is created using error-prone PCR and tolerant clones are selected under solvent stress [232]. This method has been successfully applied to construct an RNA polymerase factor-, a σ^70^- or a cAMP Receptor Protein (CRP)-based mutant collection in yeast, *Z. mobilis* or *E. coli* to improve alcohol tolerance (between 2- and 100-fold increase in survival) and/or production [13,228,233,234]. Moreover, Liang et al. combined metabolic and tolerance engineering to create a styrene-producing *E. coli* strain with superior yields. To improve styrene tolerance, the authors targeted several key regulators (including *lexA*, *narP,* and *modE*) for CRISPR editing in the pathway-optimized strain [15]. In addition to polymerases, transcriptional regulators, etc., small RNAs (sRNAs) might also be relevant targets for tolerance engineering since they are often associated with multigene networks [235]. Indeed, two sRNAs (Zms4 and Zms6) in *Z. mobilis* were recently identified as important determinants for ethanol tolerance [236]. Unfortunately, the link between these sRNAs and production has not been evaluated yet, but sRNAs might potentially provide opportunities for further strain engineering.

Finally, genome shuffling offers the possibility to create combinatorial libraries with a rich mutational diversity which cannot be obtained by rationale engineering methods (Table 2) [237]. In *S. cerevisiae*, (large-scale) genome shuffling has resulted in hybrid strains with superior fermentation traits thanks to improved ethanol tolerance [229,238]. Similarly, interspecies protoplast fusion of *S. stipitis* and *S. cerevisiae* gave rise to hybrids with superior ethanol production, compared to their parental strains [230]. In the bacterium *C. beijerinckii*, de Gérando et al. ended up with a high isopropanol-producing mutant when they combined chemical mutagenesis and genome shuffling strategies [231].

In short, boosting a microbe’s tolerance by means of rationale or global (Table 2) engineering is not only suitable for promoting microbial survival under solvent stress, but also for increasing product output.

## 6. Conclusions

End-product toxicity is a serious production bottleneck in case of microbiologically derived solvents, since these molecules have a profound impact on the survival of producer strains. Over the years, research has revealed solvent-specific defense mechanisms (e.g., efflux pumps), but also common stress responses that act against different fuels and biochemicals. Especially, this cross-compound tolerance potential might be exploited to develop a (semi)universal, tolerance engineered microbe for different production applications. In this way, redesigning a production strain at the metabolic as well as the tolerance level will ensure high productivity of the desired solvent, even at extreme product concentrations.

## Figures and Tables

**Figure 1 microorganisms-09-00249-f001:**
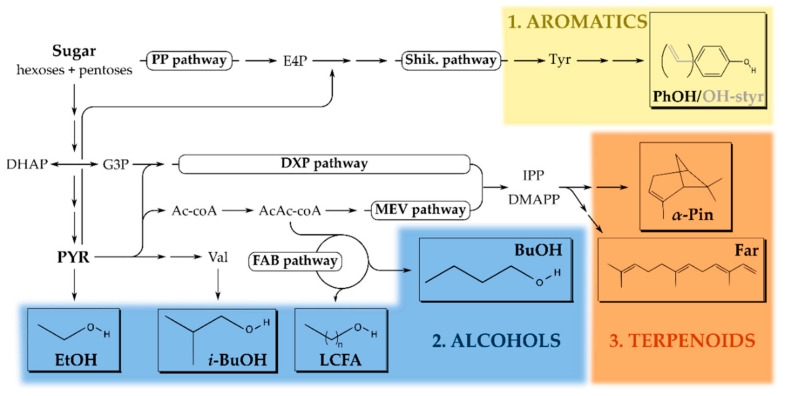
Overview of the microbial, sugar-based, solvent production pathways for alcohols, terpenoids, and aromatic compounds. (**1**) Aromatics (yellow) are predominantly based on Tyr(osine) and synthesized through the shikimate (Shik.) pathway [47,48]. (**2**) Alcohols (blue) are derived from metabolic routes such as glycolysis, fatty acid biosynthesis (FAB), and the branched amino acids pathway (such as Val(ine)) [49,50]. (**3**) Terpenoids (orange) are derived from the isopentenyl pyrophosphate (IPP) or dimethylallyl pyrophosphate (DMAPP) precursors which emerge from the 1-deoxy-D-xylulose 5-phosphate (DXP) or mevalonate (MEV) isoprenoid pathways [42,51,52]. Abbreviations: PP, pentose-phosphate; E4P, erythrose-4-phosphate; DHAP, dihydroxy-acetone phosphate; G3P, glyceraldehyde-3-phosphate; PYR, pyruvate; Ac-coA, acetyl-coenzyme A; AcAc-coA, acetoacetyl-coenzyme A; EtOH, ethanol; (*i*-)BuOH, (iso-)butanol; LCFA, long-chain fatty alcohols; α-Pin, α-pinene; Far, farnesene; PhOH, phenol and OH-styr, hydroxy-styrene.

**Figure 2 microorganisms-09-00249-f002:**
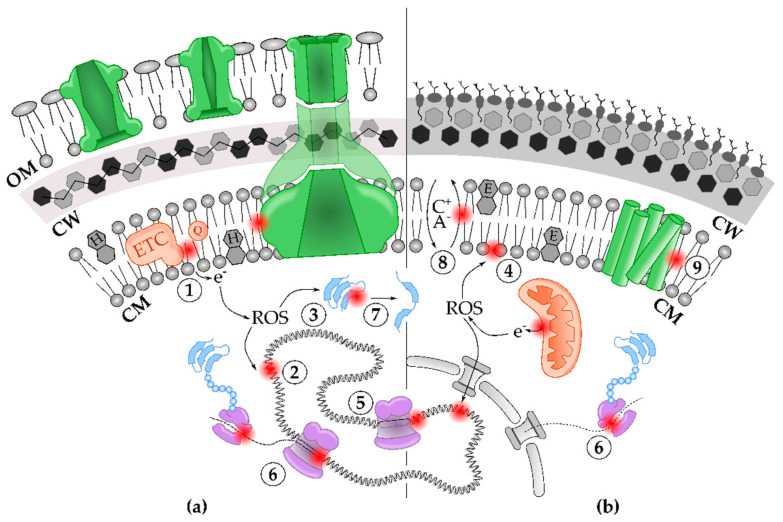
The detrimental effect of toxic solvent end-products on bacterial (**a**) and yeast (**b**) physiologies. The bacterial envelope includes (from inside to outside) the phospholipid bilayer of the cytoplasm membrane (CM), decorated with hopanoids (cyclic lipids denoted with “H”), the cell wall (CW), composed of peptidoglycan, and, in the case of Gram-negative species, the outer membrane (OM) [53]. The yeast envelope consists of a phospholipid-containing CM, including ergosterol (cyclic lipids denoted with “E”), and the chitin-rich CW [54]. (1) Solvents often induce electron leakage from electron transport chains (ETCs, orange), either situated at the inner CM in prokaryotes or in mitochondria (orange) in yeast. Eventually, these electrons give rise to reactive oxygen species (ROS) which, in turn, cause DNA damage (2), protein oxidation (3), and lipid peroxidation (4). Next, solvent molecules interfere with DNA replication (5), transcription, and translation processes (DNA, RNA polymerases and ribosomes are depicted in purple) (6). Moreover, solvent toxicity also disrupts structure and function of cytoplasmic proteins (blue) (7). Furthermore, solvents cause severe membrane damage at the phospholipid bilayer (8) which also disturbs anion (A^−^), cation (C^+^) fluxes and transport processes. Finally, these membrane deformations or direct solvent interactions also result in dysfunction of membrane-associated proteins (green) (9).

**Figure 3 microorganisms-09-00249-f003:**
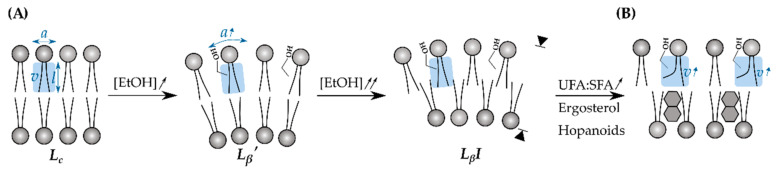
Ethanol has a profound effect on the geometry of the phospholipid membrane. (**A**) The packing geometry is defined by the phospholipid headgroup–water interface area (*a*), the hydrocarbon chain length (*l*), and the hydrocarbon chain volume (*v*) and is mathematically described as (*v*/*l*)/*a*. In absence of ethanol, the packing geometry of the membrane bilayer is between 0.5–1, which corresponds to the crystalline phase (*L_c_*). Ethanol tends to enrich at the phospholipid–water interface which significantly increases *a*. Elevated ethanol concentrations ([EtOH]) result in a micellar, disordered gel phase (*L_β_’*) and eventually cause interdigitation (*L_β_I*), in which the acyl chains of opposing monolayers are interpenetrated, leading to bilayer thinning (arrows). (**B**) This adverse configuration can be counteracted by increasing the ratio of unsaturated (UFAs)-to-saturated fatty acids (SFAs). Alternatively, ergosterols (in case of yeast) or hopanoids (in case of *Z. mobilis*) can also be incorporated in the phospholipid structure. Both response strategies help to increase *v* which (partially) restores the original packaging geometry [10,62].

**Table 1 microorganisms-09-00249-t001:** Brief summary on the bacterial (B) and yeast (S) species presented in this review.

Species	Description	Ref.
*Escherichia coli* (B)	This γ-proteobacterium is by far the most studied (bacterial) model organism. Since *E. coli* is genetically and metabolically well-characterized, its potential in the production of fuels (alcohols and terpenoids, etc.), organic acids (e.g., hydroxybutyrate), and amino acids has been explored over recent decades.	[16,17]
*Zymomonas mobilis* (B)	Originally, this α-proteobacterium was isolated from tropical, fruit-or agave-based beverages and spoiled ciders. However, its remarkable ethanol tolerance and glucose consumption rate have promoted its use in the ethanol industry. Recently, extensive metabolic engineering resulted in industrial strains which are able to produce ethanol, sorbitol, and levan from lignocellulosic biomass (composed of nonedible sugars).	[18,19,20]
*Clostridium* sp. (B)	*C. acetobutylicum* and *beijerinckii* have the natural ability to metabolize sugars into acetone, butanol and ethanol simultaneously. Therefore, these strains have been used for over 100 years to produce this solvent mix on an industrial scale.	[21,22]
*Corynebacterium glutamicum* (B)	This actinobacterium is particularly suitable for the production of amino acids. Recently, researchers have successfully implemented (biogas-based) methanol as a carbon source for this purpose.	[14,23]
Lactic Acid Bacteria (B)	This group of bacteria (including *Lactobacillus plantarum* and *Oenococcus oeni*) are naturally present in wines. These microorganisms facilitate maturation of (red) wines as they are largely responsible for the conversion of lactate into malate. The latter improves the sensory qualities and ensures that these alcoholic drinks are microbiologically stable (on the long term). As they need to withstand ethanol percentages (>10%), lactic acid bacteria are suitable candidates to study alcohol tolerance.	[24,25]
*Saccharomyces cerevisiae* (Y)	This yeast species is without any doubt the most commonly used fermentation strain both in the food and fuel industries. Decades of research on metabolic engineering even expanded the application potential of *S. cerevisiae* towards lignocellulose-based ethanol production.	[26,27]
*Kluyveromyces marxianus* (Y)	This dairy yeast is traditionally used for the fermentation of milk into yoghurt, kefir, etc. Moreover, the strain has also been industrially exploited for the production of enzymes (e.g., pectinases and lipases). Recently, researchers have also implemented *K. marxianus* in bioethanol production as this yeast displays high thermotolerance and has a broad sugar utilization range.	[28,29]
*Scheffersomyces stipitis* (Y)	This respiratory yeast is also known as *Pichia stipitis.* In contrast to *S. cerevisiae*, *S. stipitis* naturally utilizes a whole arsenal of (hemi)cellulases to consume complex sugars (e.g., cellobiose). This feature is particularly interesting in lignocellulose-based bioethanol production settings.	[30]

**Table 2 microorganisms-09-00249-t002:** Four categories of tolerance engineering strategies which improve solvent production in various microbial producers.

Strategy	Solvent	Organism	Production Gain	Reference
Adaptive laboratory evolution (ALE)	ethanol	*E. coli*	+3–16%	[219]
		*S. cerevisiae*	+20–35%	[220]
		*K. marxianus*	+120–730%	[221]
		*S. stipitis*	+10%	[223]
	butanediol	*E. coli*	+30–70%	[222]
	butanol	*C. acetobutylicum*	+44%	[12]
	methanol	*C. glutamicum*	+156%	[14]
	pinene	*E. coli*	+31%	[207]
Overexpression of stress-response pathways or detoxification mechanisms	ethanol isopentenol	*E. coli*	+11–30%	[141]
		*S. cerevisiae*	+20%	[210]
		*E. coli*	+12–60%	[156]
	butanol	*C. acetobutylicum*	+33–40%	[226]
	limonene	*E. coli*	+65%	[206]
	amorphadiene	*E. coli*	+286–308%	[208]
	phloroglucinol	*E. coli*	+39.5%	[227]
	vanillin	*S. pombe*	+25%	[192]
Global transcription machinery engineering (gTME)	ethanol	*S. cerevisiae*	+15%	[13]
		*Z. mobilis*	+90%	[228]
	styrene	*E. coli*	+31–245%	[15]
Genome shuffling	ethanol	*S. cerevisiae*	+2–7%	[229]
		*S. cerevisiae/S. stipitis*	+4–14%	[230]
	isopropanol	*C. beijerinckii*	+15%	[231]

## Data Availability

Not applicable.

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
