# Peer review of "Increasing Solvent Tolerance to Improve Microbial Production of Alcohols, Terpenoids and Aromatics"

_microorganisms, 2021, doi:10.3390/microorganisms9020249_

Round 1
Reviewer 1 Report
The review deals with an interesting and broad topic of how model microbes adjust when producing certain solvent-like molecules. Tolerance is a complicated phenotype to study and the authors did an overall great job in summarizing the current literature in this field. The paper is generally well-written but there are some word choices and sentence structuring that were suspect. To help the reader it could be useful to give at times the concentrations at which tolerance were investigated (like in 3.1).
In my view, a big part of this field is not discussed (and should be included) and that is glycosylating aromatics in order to make them less toxic. This has been done for vanillin but I am not sure of other attempts. I am including some references below. This should be included in the table and the latter part of the manuscript
http://dx.doi.org/10.1016/j.meteno.2015.09.001
DOI 10.1002/bit.24731
doi:10.1128/AEM.02681-08
Some comments:
Throughout the manuscript the field error (“Error! Reference source not found”) pops up in bold. I am not sure how it affects the referencing, but it does not seem to affect the text. Yet it definitely points out that the authors did not read the final version which they claim in line 571
Fig 1 is not mentioned in the text. Although showing the eukaryotic and prokaryotic envelope is useful here, ROS are not yet discussed which makes the placement odd.
Line 70. I guess methanol is more simple than ethanol
Line 74 not sure why iso needs to be in brackets
The text of Figure 2 is a bit fuzzy. In addition to IPP one should also include DMAPP. Labelling of farnesene is missing in the picture.
Figure 3 is not mentioned in the text. Line 145, 159 please provide the concentration values
Line 162. “Short chain” is hyphenated
Line 172. Unnecessary comma
Line 190. “Experience harm” is an odd choice of words
Line 212. “a thread to” proteins is an odd choice
Line 212. “Although” incorrectly hyphenated
Line 241: typo with “translation”
Line 244: “terpenoid-treated”
Line 301: write out “resp.”
Line 343: “contribute”
Line 361: rather should be omitted
Line 374: I’m not sure but is “abortion” often used in this context?
Line 387. Check if it is “GroeELS”. I’m sure it is not.
Line 402. Ethanol-adapted (hyphen)
Line 441: “thanks to” should be “due to”
Line 461: remove the “the”
Line 517: Table 1 is not mentioned in the text. It could also be useful if a specific outcome (hypothetical example 2-fold increase in tolerance to ethanol in LB media) would be mentioned in a separate column.
Section 5. the a), b) , c), d) in the text is clumsy and should be removed. As mentioned earlier, this section should also tackle the whole concept of glycosylation
Line 560-562. The sentence implies that these solvents affect basically everything of the cell. It is a very obvious sentence and a tedious way to conclude the review. It should be removed
Line 564. Could the authors please use another way of saying “on the one hand”? as it has been used a couple of times in the text
References lacking:
Two papers on S. cerevisiae’s response in limonene should be included as it is relevant to the topic (fyi I am not an author of either of the publications)
DOI:10.1007/s00253-013-4931-9
doi:10.1128/AEM.00463-13
Some species names like in line 372 and line 377 are not in italics
Some references are not according to the journal’s requirements e.g. line 599…every word starts with a capital letter
Reviewer 2 Report
The authors present very interesting review about.
Increasing solvent tolerance to improve microbial production 2 of alcohols, terpenoids and aromatics.
In general it is well done manuscript, but befor acceptance I recommend to add few voliable information which will strengthen this work.
Below point by point was present problem which must be changed.
- Line 71 – Authors should present most information about different specios yeast which are associated in ethanol production. S. cerevidsiae is one of the yeast. Please complete the text with other species of the genus Pichia or Kluyveromyces. Here you can also divide them into conventional and unconventional. Please remember that you are presenting data on microorganisms. This is also a Review paper, not a Mini review. Likewise with bacteria.\
- Why Authors not use cell wall term in the case bacterial cells? A cell wall is a layer located outside the cell membrane found in bacteria, but also in plants, fungi, algae, and archaea. Therefore line 113-115 must be changed and adapted to general knowledge. The presented statement may be confusing, e.g. for people who are not specialists in the field (students), but want to increase their general knowledge. I understand that this is a kind of 'shortcut'.
- There is no reference to figures 1 and 3 in the text. I understand that: 'Error! Reference source not found', it is appropriate link, or rather was.
- Paragraph 3.3. The authors write nothing about the effects of ethanol on transcription and translation. The concentration of ethanol, which changes the metabolism of the cell from anaerobic to aerobic, should be more strongly described.
- The authors described only selected types of stress, but also no other significant types of cellular stress. Please complete the text with osmotic stress, exposure to high temperature (after all, in technological processes mentioned by the authors it is an important aspect). Since the authors also mention cell viability, please describe the effect of bacterial contamination on fermentation processes and yeast culture viability. This biological stress is also essential to the overall picture of this work.
Round 2
Reviewer 1 Report
I am happy with how the authors have addressed the suggestions
Reviewer 2 Report
Well done! Congratulation.